# Gastroprotective Effect of Isoferulic Acid Derived from Foxtail Millet Bran against Ethanol-Induced Gastric Mucosal Injury by Enhancing GALNT2 Enzyme Activity

**DOI:** 10.3390/nu16132148

**Published:** 2024-07-05

**Authors:** Xiaoqin La, Xiaoting He, Jingyi Liang, Zhaoyan Zhang, Hanqing Li, Yizhi Liu, Ting Liu, Zhuoyu Li, Changxin Wu

**Affiliations:** 1Institute of Biomedical Sciences, Shanxi University, Taiyuan 030006, China; yanz8816@163.com (Z.Z.); tingliu@sxu.edu.cn (T.L.); 2Shanxi Provincial Key Laboratory of Medical Molecular Cell Biology, Shanxi University, Taiyuan 030006, China; 3School of Life Science, Shanxi University, Taiyuan 030006, China; hxt10142022@163.com (X.H.); 13903461328@139.com (H.L.); 15034187212@139.com (Y.L.); 4Institute of Biotechnology, Shanxi University, Taiyuan 030006, China; 15235329962@163.com; 5The Key Laboratory of Chemical Biology and Molecular Engineering of National Ministry of Education, Shanxi University, Taiyuan 030006, China; 6Shanxi Provincial Key Laboratory for Prevention and Treatment of Major Infectious Diseases, Taiyuan 030006, China

**Keywords:** foxtail millet, gastric mucosal injury, isoferulic acid, GALNT2, ethanol

## Abstract

Excessive alcohol consumption has led to the prevalence of gastrointestinal ailments. Alleviating gastric disorders attributed to alcohol-induced thinning of the mucus layer has centered on enhancing mucin secretion as a pivotal approach. In this study, foxtail millet bran polyphenol BPIS was divided into two components with MW < 200 D and MW > 200 D by molecular interception technology. Combined with MTT, cell morphology observation, and trypan blue staining, isoferulic acid (IFA) within the MW < 200 D fraction was determined as the effective constituent to mitigate ethanol-induced damage of gastric epithelial cells. Furthermore, a Wistar rat model with similar clinical features to alcohol-induced gastric mucosal injury was established. Then, gastric morphological observation, H&E staining, and assessments of changes in gastric hexosamine content and gastric wall binding mucus levels were carried out, and the results revealed that IFA (10 mg/Kg) significantly ameliorated alcohol-induced gastric mucosal damage. Finally, we applied techniques including Co-IP, molecular docking, and fluorescence spectroscopy and found that IFA inhibited the alcohol-induced downregulation of N-acetylgalactosamintransferase 2 (GALNT2) activity related to mucus synthesis through direct interaction with GALNT2 in gastric epithelial cells, thus promoting mucin synthesis. Our study lays a foundation for whole grain dietary intervention tailored to individuals suffering from alcoholic gastric mucosal injury.

## 1. Introduction

The elevation in living standards has propelled a rise in alcohol consumption, subsequently leading to a notable upsurge in the incidence of diseases associated with gastric mucosal damage [1]. For example, chronic alcoholics can damage the gastric mucosa and gradually develop stomach cancer [2]. Therefore, digging into the nutrients in food that have the effect of protecting the stomach and preventing the development of diseases related to gastric mucosal damage are extremely urgent.

The mucous barrier is a non-water-soluble gel-like mucus layer that is in direct contact with food in the stomach and has a thickness of 0.25–0.50 mm [3]. It protects the gastric mucosa from the damage of alcohol [4]. Chronic alcohol consumption often leads to the thinning of the mucus layer [5]. The main components of the mucus layer are mucins (MUCs), which mainly include MUC5AC secreted by mucosal surface cells and MUC6 secreted by gastric gland neck cells [6,7]. The sugar chains in MUCs account for 50% to 90% of the total molecular weight. Therefore, glycosylation is an important guarantee for the normal physiological functions of MUCs [8,9,10].

The first rate-limiting step in glycosylation is that polypeptide-N-acetylgalactosaminyl transferase (GALNTs) mediates the first glycosyl, N-acetylgalactosamine (GalNAc), and transfers it to the serine/threonine (Ser/Thr) residues of mucin peptide [11]. The structure formed by linking the first monosaccharide GalNAc to the Ser/Thr residues of protein is called the Tn antigen, which is also the basis for the glycosylation of MUCs [12]. Therefore, GALNT-mediated glycosylation plays a crucial role in maintaining the thickness and function of the mucus layer.

Foxtail millet (*Setaria italica*), a member of the Poaceae grass family, is the main cereal in northern China [13]. Traditionally, foxtail millet has an important tonic effect on gastric mucosal damage caused by alcohol consumption, but the specific active components and molecular mechanisms remain unclear. José M. Álvarez-Suárez demonstrated the efficacy of polyphenols extracted from various strawberry varieties in significantly alleviating alcohol-induced gastric mucosal damage [14]. In addition to strawberry polyphenols, tea polyphenols have also exhibited promising therapeutic effects in ameliorating gastric mucosal barrier injury induced by hydrochloric acid/ethanol in mice, further highlighting the potential of natural polyphenols in gastrointestinal protection [15,16].

In recent years, many scholars have identified that foxtail millet is rich in polyphenols, which are chiefly concentrated in the husk of foxtail millet bran [17]. We previously extracted bound polyphenol of the inner shell from foxtail millet bran (named BPIS) and found that it prevented alcohol damage to human gastric epithelial cell lines and rat gastric mucosa [18]. UPLC-triple-TOF liquid chromatography–mass spectrometry identified 12 types of polyphenol compounds in BPIS (the compounds are listed in Appendix A) [19]. Here, we demonstrate that isoferulic acid (IFA) is the main functional component in BPIS; it can improve blocked MUC synthesis by enhancing GALNT2 enzyme activity in gastric mucosal epithelial cells, maintaining the normal mucus barrier function for drinkers. Our data lay a foundation for the formulation of nutritional dietary intervention strategies for people with alcoholic gastric mucosal injury.

## 2. Materials and Methods

### 2.1. Chemicals

Foxtail millet bran was purchased from Tian-Xia-Gu Limited Company (Taiyuan, China). Isoferulic acid (IFA, Cat number: I115853, purity ≥ 98%) and 4-Hydroxybenzoic acid (4-HBA, Cat number: H117675, purity 99%) were purchased from Aladdin (Shanghai, China). Ferulic acid (FA, Cat number: F8330, purity ≥ 99%), *p*-Coumaric acid (*p*-CA, Cat number: SC8560, purity HPLC ≥ 98%), Vanillic acid (VA, Cat number: SV8020, purity HPLC ≥ 98%), Syringic acid (SA, Cat number: SS8780, purity ≥ 98%), D-Glucosamine hydrochloride (Cat number: SG8490), and Alcian Blue8GX (Cat number: A8140) were purchased from Solarbio (Beijing, China). Acetylacetone (Cat number: P7754) and 4-Dimethylaminobenzaldehyde (Cat number: 109762) were purchased from Sigma-Aldrich (St. Louis, MA, USA). The dialysis bag (Cat number: T10-03-005) was from Viskase (Chicago, IL, USA). Antibodies for MUC5AC (Cat number: ab198294), MUC6 (Cat number: ab192318), GALNT1 (Cat number: ab253025), GALNT2 (Cat number: ab262868), GALNT10 (Cat number: ab106471), GALNT12 (Cat number: ab101358), GALNT14 (Cat number: ab235526), and GAPDH (Cat number: ab8245) were from Abcam (Cambridge, UK). GALNT15 antibody (Cat number: PA5-113152) was obtained from Invitrogen (Carlsbad, CA, USA). Recombinant human MUC5AC protein (Cat number: ab114218) and recombinant human GALNT2 protein (rGALNT2) (Cat number: ab276585) were from Abcam (Cambridge, UK). The recombinant human MUC6 protein (Cat number: RPB489Hu01) was from Cloud-Clone (Wuhan, China).

### 2.2. BPIS Extraction and Component Separation

The extraction of BPIS was performed as previously described [19]. Briefly, foxtail millet bran was initially dissolved in pre-chilled acetone, followed by stirring and subsequent centrifugation. The resultant sediment was treated with sodium hydroxide, adjusting the pH to neutrality at 7, after which, it was centrifuged again to collect the supernatant. This collected supernatant then underwent a refined process including defatting, rotary evaporation for solvent removal, dialysis to purify, and, ultimately, freeze-drying, yielding the desired BPIS.

MS analysis identified twelve polyphenolic compounds in BPIS (Appendix A). These compounds were categorized based on their molecular weight, with six substances exceeding 200 D and another six falling below this threshold. So, we used 200 D as the dividing line to divide the BPIS into two groups for the experiments. In this study, a dialysis bag with a molecular mass cut-off of 200 D was used to separate BPIS. After stirring and dialysis at 4 °C for 36 h, BPIS in the dialysis bag (MW > 200 D) and dialysate (MW < 200 D) were concentrated by rotary evaporation.

### 2.3. Cell Experiment

#### 2.3.1. Cell Line and Cell Culture

Human gastric mucosal epithelial cells (GES-1) were supplied by Zhejiang Noble Biological Products Co., Ltd. (Hangzhou, China). These cells were cultured in RPMI-1640 medium supplemented with 10% FBS at 37 °C in a humidified tissue culture incubator containing 5% CO_2_.

#### 2.3.2. Protective Effects of Different Components of BPIS on Ethanol-Induced Damaged Cells

GES-1 cells were treated with 1000 mM EtOH for 12 h to establish an ethanol-induced gastric epithelial cell injury model according to our previous report [18]. GES-1 cells were divided into three groups: (a) control group; (b) model group; and (c) different-dose BPIS components groups.

For MTT assays, cells were seeded in 96-well plates for 12 h before the drug was added. After 24 h of different-dose BPIS components treatment, all cells except for those of the control group were treated with 1000 mM EtOH for another 12 h. The control group was treated with an equivalent volume of phosphate-balanced solution (PBS) for 12 h. Subsequently, MTT assays were performed according to our previous procedure [20].

For trypan blue stain assays, cells were seeded in 12-well plates and treated with the same protocol as in the MTT experiment above. Subsequently, the cells were stained with trypan blue and photographed with a microscope.

### 2.4. Animal Experiment

#### 2.4.1. Animal Experiment Design

Male Wistar rats (9–10 weeks old, *n* = 50) were purchased from Beijing Vital River Laboratory Animal Technology Co., Ltd. (Beijing, China). The rats were acclimatized to an SPF-grade environment for 7 d before being used for this study. Fifty rats were randomly divided into 5 groups, with 10 animals per group as follows: (a) control group; (b) IFA alone treatment group; (c) EtOH group; (d) IFA intervention group: gavaged with 5 mg/kg/day IFA (IFA_Low_); and (e) IFA intervention group: gavaged with 10 mg/kg/day IFA (IFA_High_). The control group and EtOH group were gavaged with 1 mL/100 g/day of physiological saline. After 21 successive days of oral administration, all rats fasted for 24 h with free access to food. All the rats except for those in the control group and IFA alone treatment group were given 75% EtOH (5 mL/kg); 4 h later, the rats were sacrificed after anesthesia with 10% chloral hydrate. All animal experiments conducted in this study were approved by the Institutional Animal Care and Use Committee of Shanxi University (Taiyuan, China). All of the experimental procedures were performed in accordance with the protocols and ethical regulations approved by the Institutional Animal Care and Use Committee of Shanxi University (Taiyuan, China).

#### 2.4.2. Macroscopic Assessment of the Gastric Mucosal Injury and Guth Injury Index Score

The rats were cut along the midline of the abdominal cavity and the gastric tissue was removed and opened along the greater curvature of the stomach. The contents of the stomach were washed in precooled physiological saline and then photographed for macroscopic assessment of the gastric mucosal injury.

The measurement of the gastric mucosal injury index was based on the Guth scoring standard: the gastric mucosal injury site of each rat was measured with an electronic caliper, and the addition of sheet injury and strip injury was the gastric mucosal injury index of this rat (Appendix A).

#### 2.4.3. H&E Staining and Pathological Index Score

After H&E staining, the gastric tissues were observed and photographed under a microscope. The pathological injury index of rat gastric mucosa was calculated according to the Mascuda standard (Appendix A). The cumulative damage score of each paraffin section should not have exceeded 15 points.

#### 2.4.4. Determination of Aminohexose and Gastric Wall-Binding Mucus

The gastric mucosal hexosamine content was estimated using the method by Feng et al. [21]. The fresh gastric mucosal tissue and mucus were gently scraped off the glandular stomach and weighed. They were then centrifuged at 10,000 rpm for 10 min after being mixed with ice physiological saline. The supernatant was reacted with acetyl acetone and 4-dimethylaminobenzaldehyde to calculate the concentration of aminohexose, while the concentration of protein in the supernatant was determined using the Coomassie brilliant blue method (mg/L). The final result was expressed as the amount of aminohexose per milligram of protein (g/mg protein).

The mucus content in gastric tissue was estimated as described by Corne et al. [22]. The glandular segment of the stomach was separated from the lumen of the stomach, weighed, and transferred immediately to 20 mL of 0.02% *w*/*v* Alcian blue solution (with 0.12 mol/L Na_2_HPO_4_ and 0.04 mol/L citric acid in a pH 5.8 buffer solution). The tissue was stained for 2 h in Alcian blue. Then, 7 mL of incubation solution was sucked and centrifuged at 3000 rpm for 20 min, and the absorbance of the aqueous layer was recorded at 580 nm. The binding mucus content of the gastric wall was calculated as the amount of bound dye according to the following formula: 4 − (20 × measured optical density/standard tube optical density × 5) = amount of combined dye (mg); 4 is the milligrams of Alcian blue contained in 20 mL of dye.

#### 2.4.5. RNA Extraction, Reverse Transcription, and Real-Time PCR Analysis

Total RNA extraction, reverse transcription, and real-time PCR analysis were performed as previously described [20]. Briefly, to extract the total RNA, Trizol was employed. The concentration of RNA was determined using a Thermo Scientific Nanodrop-2000. cDNA synthesis was carried out through reverse transcription, involving incubation at 37 °C for 15 min followed by activation at 85 °C for 5 min. For the precise quantification of gene expression, qRT-PCR was conducted using SYBR Green chemistry, with all samples being assessed in triplicate. The primer sequences used in this study are listed in Appendix A.

#### 2.4.6. *Vicia villosa* Lectin Immunoprecipitation

Gastric tissues from different groups of rats were collected and the cells were lysed. The mucin Tn antigen structure was immunoprecipitated using vicia villosa lectin (VVL) [23]. Subsequently, the expression of mucin-related Tn antigen was detected using MUC5AC and MUC6 monoclonal antibodies.

#### 2.4.7. Deglycosylation of Mucins

The sugar chain of mucin was removed by periodate oxidation-β elimination during the process of extracting gastric mucosal tissue proteins. Then, 2 mg/mL protein was dissolved in a 50 mmol/L sodium acetate buffer solution at pH 4.6, and 10 mmol/L sodium periodate was added for 15 min. Next, 50 mmol/L sodium borohydride was added to terminate the oxidation reaction. After washing several times using centrifugal ultrafiltration, the deglycosylation mucins were obtained.

### 2.5. GALNT–Transferase Activity Assays

GES-1 cells were ultrasonically lysed, cell debris was removed by centrifugation, and the protein concentration in the supernatant was determined to keep the same amount of protein loaded for each group. GALNT1 (GALNT2, GALNT10, GALNT12, GALNT14, or GALNT15) antibody was added to the supernatant of the cell lysate at a final concentration of 2 μg/mL, which was incubated overnight at 4 °C with rotation. Then, 50 μL protein A + G agarose beads were supplemented into the reaction system. Next, 2 h later, the supernatant was discarded. Deposits were added to 20 μL of total reaction mixture containing 25 mM Tris-HCl (pH 7.4), 5 mM MnCl_2_, 0.25 mM UDP-[^14^C]GalNAc (1–3000 cpm/nmol), and 25 μM acceptor peptide. [^14^C]GalNAc incorporation in acceptor peptides, catalyzed by GALNTs, was determined by scintillation counting following Dowex-1 formic acid chromatography [24].

### 2.6. Molecular Docking of IFA to GALNTs

Crystal structures of GALNTs were retrieved from UniProt (https://www.uniprot.org/ accessed on 7 November 2023). The 2D structure of IFA was obtained in the PubChem database (https://pubchem.ncbi.nlm.nih.gov/ accessed on 7 November 2023). Molecular docking was performed as previously described [25].

### 2.7. Determination of the Interaction between IFA and GALNT2

Human recombinant GALNT2 protein (rGALNT2) and the lysate of GES-1 cells were incubated with IFA-Fe_3_O_4_ beads overnight at 4 °C, respectively. Uncoupled-IFA Fe_3_O_4_ beads were used as negative controls for the pull-down experiment.

### 2.8. Statistical Analysis

All experiments were performed in triplicate, and a representative experiment was selected for presentation. Statistical analysis was performed using GraphPad Prism 9.5.0. Software. Data were expressed as the mean ± SEM. Multiple comparisons among groups were performed using one-way analysis of variance (ANOVA) with posthoc Tukey tests and comparisons between two groups were evaluated using the Student *t*-test. The values of *p* < 0.05, *p* < 0.01, *p* < 0.001, and *p* < 0.0001 were considered statistically significant.

## 3. Results

### 3.1. BPIS (MW < 200 D) Significantly Alleviates Alcohol-Induced Gastric Epithelial Cell Injury

We have previously reported that BPIS can significantly alleviate alcohol-induced gastric mucosal injury in rats and GES-1 cells [18]. To explore the active components of BPIS that play a protective role in gastric mucosa, we divided BPIS by molecular mass cut-off 200 D, and then those two components were used to intervene in an alcohol-induced gastric epithelial cell injury model. The results of MTT assays showed that the MW < 200 D component could facilitate the callback of the cell viability of GES-1, in place of the MW > 200 D component (Figure 1A,B). The cell morphology experiment also showed that GES-1 was present with a normal epithelial cell-like appearance and firmer adherence after the intervention with the MW < 200 D component, while the MW > 200 D component made GES-1cells rounder and tended to be suspended, which was similar to the EtOH treatment group (Figure 1C). The trypan blue staining experiment also showed that some cells floated in PBS during the process of washing the residual dye solution, while most of the remaining adherent cells appeared blue in the intervention group of MW > 200 D. However, for the cells treated with MW < 200 D fractions, the number of cells stained with blue and the staining area were contracted (Figure 1D). These results indicated that MW < 200 D was the main active component of BPIS in alleviating alcohol-induced gastric epithelial cell injury.

### 3.2. IFA Is the Main Component of BPIS (MW < 200 D) in Inhibiting Alcoholic Gastric Mucosa Injury

BPIS (MW < 200 D) is mainly composed of six phenolic compounds; we focused on which compound was the active component to alleviate alcohol-induced gastric mucosal injury. The MTT assay was performed with six components in different concentrations, and it was found that IFA reverted the cell viability of GES-1 (Figure 2A). However, FA, VA, *p*-CA, SA, and 4-HBA could not sustain the manifestation (Figure 2B–F). The results of trypan blue staining showed that fewer cells were stained with IFA intervention, while numerous cells were stained in the other five phenolic acid treatment groups (Figure 2G). Similarly, cell morphology experiments verified that the cells in the IFA treatment group had complete structures compared with the other five phenolic acid treatment groups (Figure 3), suggesting that IFA was the main component of BPIS in improving alcoholic gastric epithelial cell injury.

### 3.3. IFA Improves Alcoholic Gastric Mucosa Injury in Wistar Rats

To determine the role of IFA, Wistar rats were used to establish an alcohol-induced gastric mucosal injury model. The results showed that the gastric mucosal surface of rats with complete and smooth folds had more mucus in the control group and IFA alone treatment group. In the alcohol treatment group, gastric mucosa showed hemorrhage, edema, and multiple linear or circular erosions of different sizes, with flat and shallow folds or even defects, with less mucous on the surface. Compared with the alcohol group, the degree of gastric mucosal bleeding and erosion was significantly reduced in the pretreatment groups with IFA, and there was more mucus (Figure 4A). The Guth scoring standard was employed to evaluate the gastric mucosal damage index. The results showed that the average gastric mucosal damage indexes of the control group, IFA alone treatment group, alcohol group, and low-concentration and high-concentration IFA pretreatment groups were 0, 0, 104.8, 30.9, and 6.8 points, respectively (Figure 4B).

H&E staining of gastric tissue showed that the gastric epithelium structure of rats was intact and the cells were arranged in an orderly manner in the control group and the IFA group. In the EtOH group, the gastric gland cells were necrotic; the capillary hyperemia and dilatation and the water quality were changed in both the mucosa and submucosa. In the pretreatment groups with IFA, although a small number of epithelial cells were occasionally shed, the structure of the mucosa, submucosa, and myometria was intact and continuous, and the glands were closely arranged (Figure 4C). The Mascuda scoring standard was used to evaluate the gastric mucosal tissue damage index, and the results showed that the average gastric mucosal tissue damage indexes of the control group, IFA alone treatment group, alcohol group, and low-concentration and high-concentration IFA pretreatment groups were 0, 0, 8.8, 2.0, and 0.2, respectively (Figure 4D). The results showed that IFA could significantly improve gastric mucosa injury caused by alcohol.

### 3.4. IFA Promotes the Expression of Mucin in the Gastric Mucosa of Wistar Rats

Hexosamine and gastric wall-binding mucus can evaluate the function of the gastric mucosal mucus barrier. We collected gastric tissue and found that compared with the control group, the contents of both for rats in the alcohol-treated group were reduced by 57% and 40%, respectively. The contents of both could be significantly increased after intervention with IFA (Figure 5A,B). Next, MUC5AC and MUC6, mucin-related genes, were detected by qRT-PCR. The results showed that there was no measurable change in the mRNA levels of the two mucins (Figure 5C,D). Furthermore, the expression of the Tn antigen was detected by immunoprecipitation with vicia villosa lectin. It was found that the expression level of the Tn antigen was significantly reduced after ethanol gavage, while this phenomenon was reversed by the intervention of IFA (Figure 5E). However, when the sugar chain was removed by the periodate oxidation-β elimination method in the process of protein extraction, the expression of *MUC5AC* and *MUC6* proteins showed no significant changes (Figure 5F). All these results indicated that IFA promoted the synthesis of mucin by increasing glycosylation modification.

### 3.5. IFA Facilitates Mucin Synthesis by Increasing the Activity of Glycosyltransferase GALNT2

To verify which glycosyltransferase mainly mediates the glycosylation modification of mucins, the gastric mucosa tissues of Wistar rats were collected, and the protein expression levels of six key glycosyltransferase GALNTs (GALNT1, GALNT2, GALNT10, GALNT12, GALNT14, GALNT15) that are widely distributed in gastric epithelium and systemic tissues were detected. The result showed that there was no obvious change in the expression of these GALNTs (Figure 6A). Subsequently, the GES-1 cell injury model was induced by 1000 mM ethanol followed by intervention with GALNTs, ^14^C-labeled GalNAc, and two receptor naked peptides, MUC5AC and MUC6, to detect the activity of GALNTs. It was found that the activity of GALNT2 decreased most remarkably when triggered by ethanol, while IFA could notably increase the activity of GALNT2 (Figure 6B,C). These results showed that IFA promoted mucin synthesis by increasing the activity of glycosyltransferase GALNT2.

### 3.6. IFA Alleviates the Inhibitory Effect of Alcohol on GALNT2 Activity by Interacting with GALNT2

How IFA affected GALNT2 activity was the next issue we were concerned about. GALNTs consist of two domains: a catalytic domain and a lectin domain. To assess the possible binding of IFA to GALNTs, we used computer molecular simulation docking technology. Then, the optimal binding mode of IFA was derived through docking to each possible binding site. The predicted binding mode of IFA to the catalytic domain of GALNT2 revealed a good shape match between IFA and the binding pocket; residues Gly333 and Asn335 of GALNT2 were involved in forming hydrogen bonds with IFA. The docking energy was higher in the pocket of GALNT2 (−6.2 kcal/mol) than in other GALNTs, indicating a strong interaction between IFA and GALNT2 (Figure 6D and Table 1). To validate the binding of IFA to GALNT2, IFA was coupled to Fe_3_O_4_ magnetic nanoparticles (IFA-Fe_3_O_4_), and we carried out a pull-down assay with recombinant GALNT2 (rGALNT2) and endogenous GALNT2 of GES-1 cells. As expected, IFA could interact with purified rGALNT2 and bind with the endogenous GALNT2 of GES-1 cells, whereas the Fe_3_O_4_ control showed no detectable interaction with GALNT2 (Figure 6E). Taken together, these results suggested that IFA could improve GALNT2 activity by directly interacting with it in gastric mucosal epithelial cells, ultimately ameliorating the impediment of mucin synthesis triggered by ethanol.

## 4. Discussion

In this study, we harnessed molecular interception technology to divide BPIS into two distinct fractions based on a molecular weight threshold of 200 D. Subsequently, through the application of MTT assays and trypan blue staining, we identified that IFA served as an efficacious component in alleviating ethanol-induced damage to gastric epithelial cells. Building upon these findings, our in vivo animal experiments further attested to IFA’s capacity to ameliorate alcohol-induced gastric mucosal injury in rats. Finally, employing a suite of techniques including Co-IP, molecular docking, and fluorescence spectroscopy, we delved deeper into the intricate mechanism by which IFA operates. Our investigations revealed that IFA directly interacts with GALNT2 within gastric epithelial cells. This interaction effectively curbs the alcohol-induced downregulation of GALNT2 activity associated with mucin synthesis, thereby stimulating the production of protective mucus (Figure 7).

At present, triple therapy is based on the synergistic treatment of gastric mucosal injury [26]. However, these drugs cannot repair the damaged tissue air [27]. The repairment of gastric mucosal damage requires the self-secretion of mucin to form a new mucus barrier. However, in some patients with weak self-repair function, the synthesis and secretion of mucin are hindered, resulting in the formation of gastritis in the gastric mucosa damaged by alcohol stimulation. Our study found that the active ingredient in millet polyphenols, IFA, can improve the alcohol-induced obstruction of mucin secretion. This discovery not only enriches our understanding of IFA’s biological effects but also opens up novel perspectives and potential strategies for the prevention and treatment of alcohol-induced gastric diseases.

Substantial evidence indicates that plant polyphenols can prevent alcohol from damaging gastric mucosa epithelial cell lines and gastric mucosa [14,28,29]. We also found that BPIS can alleviate the oxidative damage of gastric mucosa epithelial cells caused by alcohol by reducing the production of reactive oxygen species and inhibiting lipid peroxidation [18]. In this study, we identified the effective ingredient that protects gastric mucosa in BPIS. Mariza Abreu Miranda et al. found IFA in total hydroethanolic crude extract (ESC) from the leaves of *Solanum cernuum* Vell with the capacity to alleviate alcohol-induced gastric mucosal damage [30]. What sets our research apart is that while ESC exerts its protective effects by activating the body’s intrinsic antioxidant defense systems to shield the stomach from alcohol, our study zeroed in on the metabolic pathways of alcohol. We elucidated how IFA directly interacts with GALNT2 in gastric epithelial cells, reversing the decline in GALNT2 activity caused by ethanol. This interaction facilitates the restoration of mucin synthesis, alleviating the blockage and preserving the normal function of the mucus barrier, thus addressing the root cause of alcohol-induced gastric mucosal injury at its very source.

Post-translational acetylation is an important molecular regulatory mechanism affecting the biological activity of GALNTs [31]. Natacha Zlocowski acetylated purified recombinant GALNT2 with acetic anhydride and then measured the glycosyltransferase activity of GALNT2 using mucins nude peptide as a receptor peptide and found that the specific glycosyltransferase activity of GALNT2 was reduced by 95% by acetylation. Further precise identification by molecular docking technology combined with mass spectrometry showed that five acetylation sites (K103, S109, K111, K363, and S373) located in the catalytic domain were the key sites affecting the activity of glycosyltransferase [24]. The main metabolic pathway of alcohol is oxidation to acetaldehyde by alcohol dehydrogenase (ADH) and then further conversion to acetic acid by acetaldehyde dehydrogenase (ALDH). Acetyl-CoA synthetase (ACS) in cytoplasm finally activates acetic acid molecules to acetyl-CoA (Ac-CoA). The acetyl group of acetyl CoA is the only donor of acetylation modification. Therefore, Ac-CoA produced in the stomach by alcohol intake is an important factor affecting the acetylation modification of GALNT2. In the present study, we demonstrated that ethanol significantly inhibited GALNT2 activity, while IFA remarkably reversed this phenomenon. IFA interacted directly with GALNT2. Based on these results and previous findings, we speculated that IFA may inhibit alcohol-mediated acetylation modification by directly interacting with GALNT2 in gastric mucosal epithelial cells, thereby improving the obstruction of mucin synthesis. However, the specific sites of the interaction between IFA and GALNT2, as well as the mechanism of their binding to the acetylation regulation of GALNT2, are still not clear, which is our next research direction.

In summary, the gastroprotective effect of IFA derived from foxtail millet bran against ethanol-induced gastric mucosal injury was by enhancing GALNT2 enzyme activity. This study explored the molecular mechanism of bran polyphenols against gastric mucosal injury and emphasized the advantages of bran polyphenols applied as functional foods and dietary supplements for drinkers.

## 5. Conclusions

In conclusion, employing molecular interception technology in conjunction with MTT and trypan blue staining assays, we identified IFA as the active component in BPIS that exerts beneficial effects on gastric mucosa. The protective action of IFA on gastric mucosa was subsequently corroborated in vivo in a rat model with gastric mucosal damage. Further elucidation of IFA’s mechanism of action was achieved through Co-IP experiments and molecular docking studies.

## Figures and Tables

**Figure 1 nutrients-16-02148-f001:**
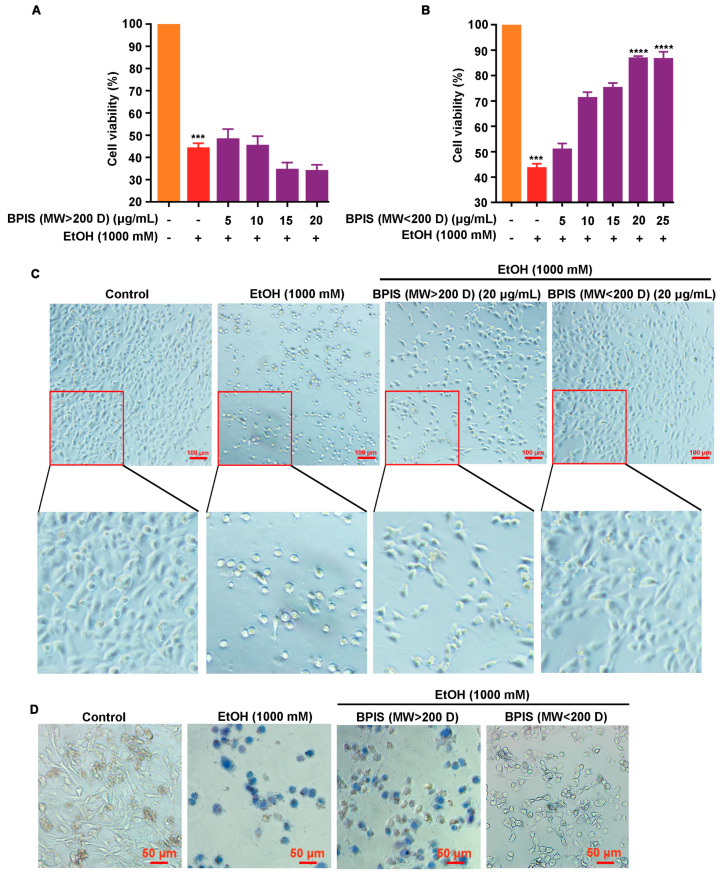
Effect of BPIS (MW > 200 D) and BPIS (MW < 200 D) components on alcohol-induced gastric epithelial cell injury. (**A**,**B**) GES-1 cells were pre-treated with different concentrations of BPIS (MW > 200 D) (**A**) and BPIS (MW < 200 D) (**B**) for 24 h and then treated with 1000 mM alcohol for another 12 h. Cell viability was detected by MTT. The column of orange in both (**A**,**B**) indicates the control group, with the alcohol-treated group in red. The purple column in (**A**) indicates the alcohol-treated intervention group with different concentrations of BPIS (>200 D), and the purple column in (**B**) indicates the alcohol-treated intervention group with different concentrations of BPIS (<200 D). *** *p* < 0.001, **** *p* < 0.0001. (**C**) Effect of MW > 200 D and MW < 200 D fractions in BPIS on the morphology of gastric epithelial cells. Scale: 100 μm. (**D**) The effect of MW > 200 D and MW < 200 D fractions in BPIS on the activity of gastric epithelial cells was detected by trypan blue staining assay. Scale: 50 μm.

**Figure 2 nutrients-16-02148-f002:**
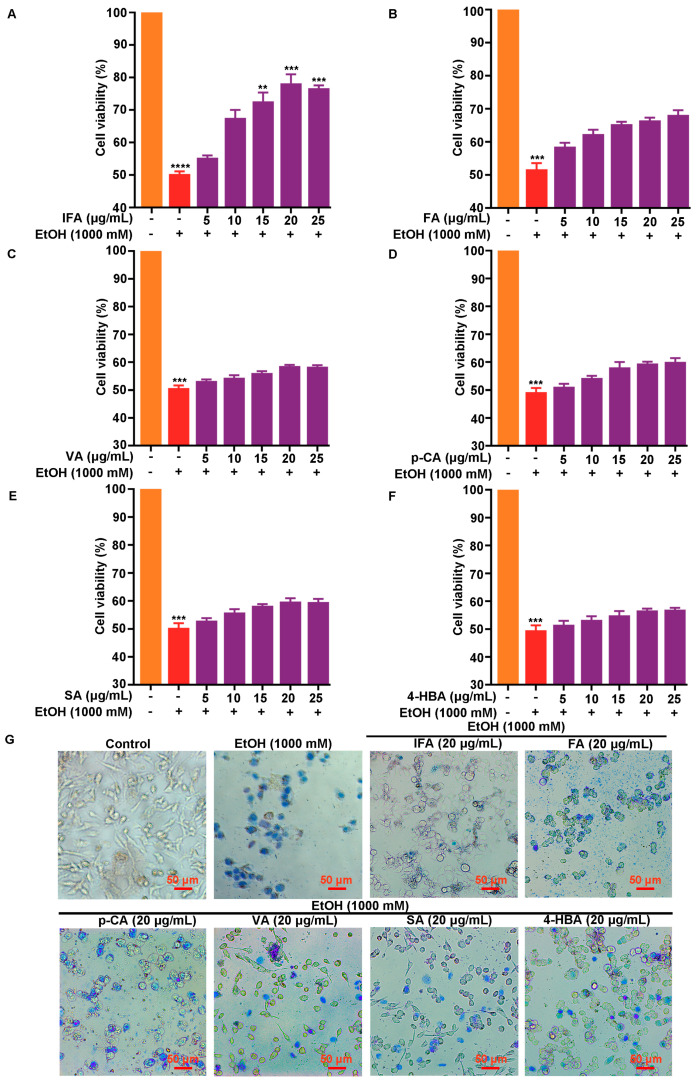
Effects of six single components in BPIS (MW < 200 D) on alcoholic gastric epithelial cell injury. (**A**–**F**) GES-1 cells were pre-treated with different concentrations of isoferulic acid (IFA), ferulic acid (FA), vanillic acid (VA), *p*-coumaric acid (*p*-CA), syringic acid (SA), and 4-hydroxybenzoic acid (4-HBA) for 24 h and then treated with 1000 mM alcohol for 12 h. Cell viability was detected by MTT. The orange columns in (**A**–**F**) all indicate the control group, the red columns all indicate the alcohol-treated group and the purple columns indicate the different concentrations of IFA, FA, VA, p-CA, SA, and 4-HBA, respectively. (**G**) The effects of six single components in BPIS (MW < 200 D) on gastric epithelial cell activity were analyzed by trypan blue staining assay. Scale: 50 μm. ** *p* < 0.01, *** *p* < 0.001, **** *p* < 0.0001.

**Figure 3 nutrients-16-02148-f003:**
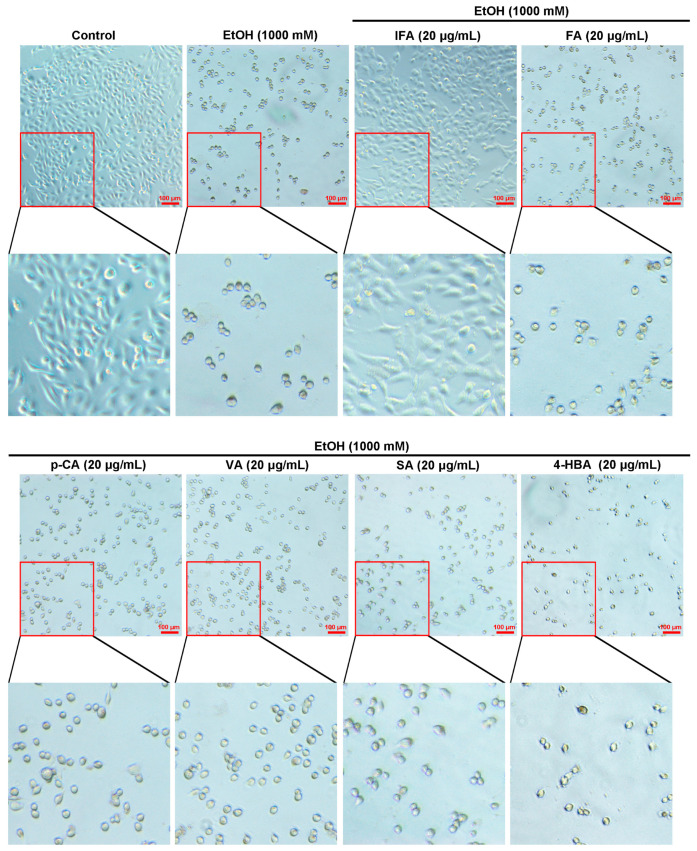
Effects of six single components in BPIS (MW < 200 D) on the morphology of alleviating alcoholic gastric epithelial cell injury. Light micrographs were exhibited in GES-1 cells. The cells were pre-treated with different concentrations of BPIS (MW < 200 D) components for 24 h and then treated with 1000 mM alcohol for another 12 h. Representative images from three independent experiments are shown above. Scale: 100 μm.

**Figure 4 nutrients-16-02148-f004:**
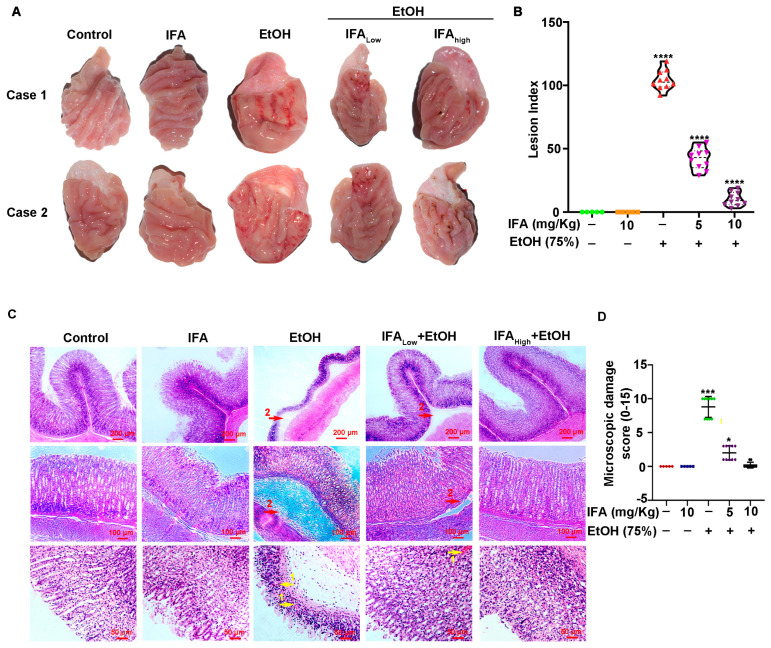
Effects of IFA on alcohol-induced gastric mucosal injury in Wistar rats. (**A**) Representative images of gastric tissue in different groups. (**B**) Statistical graph of Guth index score of gastric tissue. Green indicates the control group, orange indicates the IFA-only intervention group, red indicates the alcohol-treated group, light purple indicates the alcohol-treated low-dose IFA intervention group and dark purple indicates the alcohol-treated high-dose IFA intervention group. **** *p* < 0.0001. (**C**) Representative H&E staining images of gastric tissue. Arrow 1 in yellow indicates inflammatory cell infiltration, arrow 2 in red indicates edema. Scale: 200 μm, 100 μm, 50 μm. (**D**) Statistical map of Mascuda injury index score of H&E staining of gastric tissue in different groups. Red indicates the control group, blue indicates the IFA-only intervention group, green indicates the alcohol-treated group, dark purple indicates the alcohol-treated low-dose IFA intervention group, and black indicates the alcohol-treated high-dose IFA intervention group. * *p* < 0.5, *** *p* < 0.001.

**Figure 5 nutrients-16-02148-f005:**
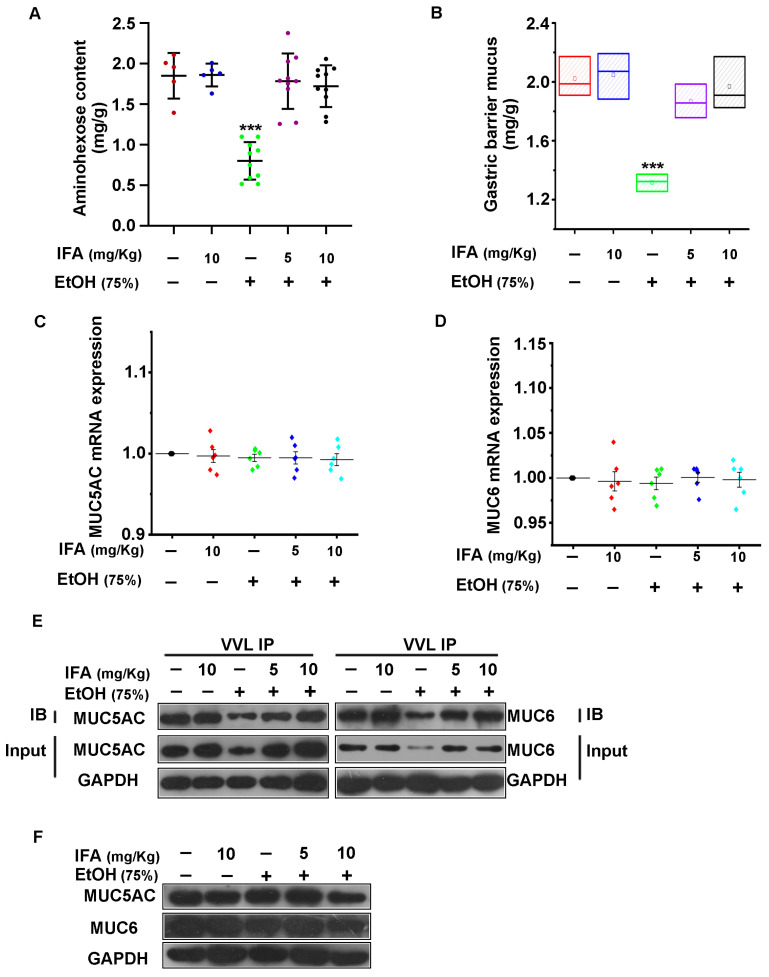
Effects of IFA on mucin expression in gastric mucosa. (**A**,**B**) Statistical map of hexosamine and adhesive mucus content in gastric mucosa. Red represents the control group, blue represents the IFA-only treatment group, green represents the alcohol treatment group, purple represents the alcohol-treated and low-dose IFA treatment group, and black represents the alcohol and high-dose IFA treatment group. *** *p* < 0.001. (**C**,**D**) Real-time PCR was used to detect MUC5A and MUC6 mRNA levels in different gastric mucosa tissues. Black represents the control group, red represents the IFA-only treatment group, green represents the alcohol-treated group, dark blue represents the alcohol-treated and low-dose IFA-treated group, and light blue represents the alcohol-treated and high-dose IFA-treated group. (**E**) Co-IP assay was used to detect the change in mucin Tn antigen expression. (**F**) Western blot assay was performed to detect MUC5AC and MUC6 expression levels after the sugar chain of mucin was removed by periodate oxidation-β elimination during the process of extracting gastric mucosal tissue proteins.

**Figure 6 nutrients-16-02148-f006:**
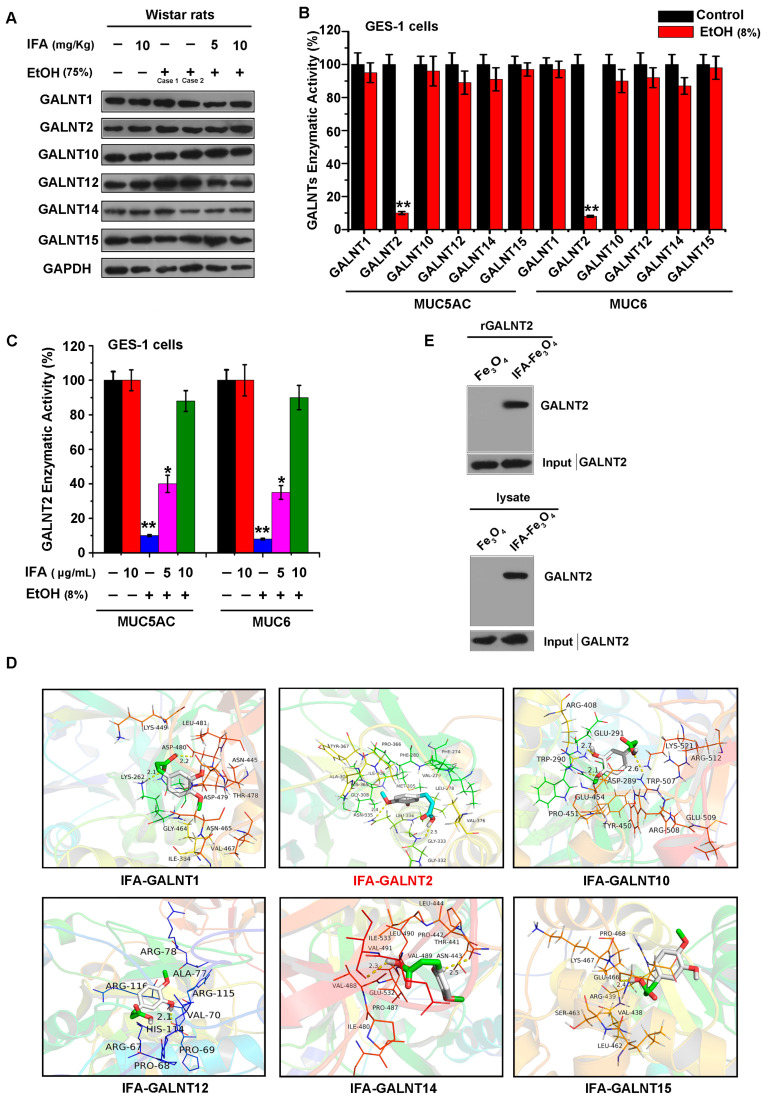
Effects of IFA on GALNTs-related gene expression and enzyme activity. (**A**) Western blot was used to detect the effect of alcohol and IFA pretreatment on the expression levels of GALNTs in the gastric mucosal tissues of rats. (**B**,**C**) Determination of GALNT glycosyltransferase activity in GES-1 cells. The color of the control group in black and the alcohol-treated group in red (**B**). The enzyme activity of GALNTs under different conditions was measured using receptor peptides MUC5AC and MUC6. Black represents the control group, red represents the IFA-only treatment group, blue represents the alcohol treatment group, purple represents the alcohol-treated and low-dose IFA treatment group, and green represents the alcohol and high-dose IFA treatment group (**C**). * *p* < 0.05, ** *p* < 0.01. (**D**) Molecular docking results of IFA interaction with GALNT1, GALNT2, GALNT10, GALNT12, GALNT14, and GALNT15. (**E**) Pull-down experiment was utilized to detect the interaction between IFA and GALNT2. rGALNT2 (upper image) and GES-1 cell lysate (lower image) were incubated with IFA-Fe_3_O_4_ coupled magnetic beads overnight at 4 °C, and the interaction was detected by GALNT2 antibody.

**Figure 7 nutrients-16-02148-f007:**
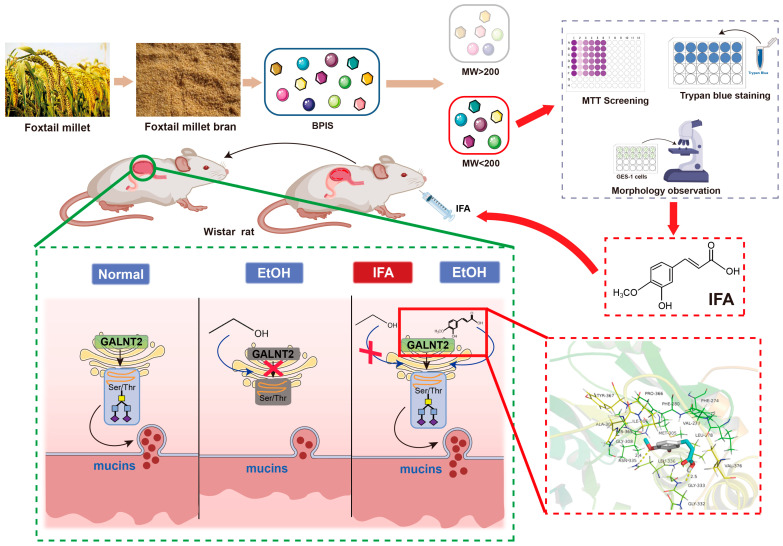
Study of the mechanism of alleviating ethanol-induced gastric mucosal injury by IFA derived from foxtail millet bran. Isoferulic acid (IFA), the active component of foxtail millet, plays a crucial role in ameliorating alcohol-induced gastric mucosal injury. IFA inhibits the decrease in glycosyltransferase GALNT2 activity caused by alcohol by directly interacting with GALNT2 in gastric epithelial cells, consequently improving the blocked synthesis of mucus and maintaining normal mucus barrier function.

**Table 1 nutrients-16-02148-t001:** The results of molecular docking for IFA with GALNT1, GALNT2, GALNT10, GALNT12, GALNT14, and GALNT15.

Protein Name	NCBI Gene Number	UniProtKB/Swiss-Prot Number	Residue Involved in H-Bonding	H-Bond Length (A)	Binding Energy (kcal/mol)
GALNT1	2589	Q10472	LYS262; ASP480	2.1; 2.2	−5.7
GALNT2	2590	Q10471	GLY333; ASN335	2.5; 2.4	−6.2
GALNT10	55568	Q86SR1	TRP290; ARG408; ARG508	2.1; 2.7; 2.6	−5.3
GALNT12	79695	Q8IXK2	HIS114	2.1	−5.2
GALNT14	79623	Q96FL9	ASN443; ILE533	2.5; 2.3	−4.9
GALNT15	117248	Q8N3T1	GLU466	2.4	−5.0

## Data Availability

The datasets used and analyzed during the current study are available from the corresponding author upon reasonable request. The data are not made public for privacy reasons.

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
