# Peer review of "Gastroprotective Effect of Isoferulic Acid Derived from Foxtail Millet Bran against Ethanol-Induced Gastric Mucosal Injury by Enhancing GALNT2 Enzyme Activity"

_nutrients, 2024, doi:10.3390/nu16132148_

Round 1

Reviewer 1 Report

Comments and Suggestions for Authors

The authors considered the foxtail millet bran polyphenol BPIS, split into two components with MW<200D and MW>200D, as an effective ingredient in mitigating ethanol-induced gastric epithelial cell damage in a Wistar rat model. In addition, IFA inhibited the alcohol-induced downregulation of N-acetylgalactosaminetransferase 2 (GALNT2) activity associated with mucus synthesis by directly interacting with stomach GALNT2. The manuscript is well structured, written in standard English with some minor phraseological remarks.

Small comments

1. With the improvement in living standards...- to change

2. on page 2 - remove excess space

3. page 2 - unifying the font...line 52-60

4. table 1 and 2 are boring - present them anew; explanations are insufficient

5. on page 4 - remove excess space

6.2.4.1. Animals and 2.4.2. Animal Experimental Design - to unite

7.2.4.6. RNA extraction, Reverse transcription, and Real-time PCR analysis- the entire paragraph should be redone

8. page 9 to fix - too much empty space

9. the caption of table 4 to be unified

10. due to the importance of the study, the discussion should be revised and enriched (graphic version of the probable mechanism of action of BPIS/ MW<200D and MW>200D)

11.only 6 (7%) used references are from the last 4 years

Comments on the Quality of English Language

-

Reviewer 2 Report

Comments and Suggestions for Authors

The authors investigate the gastroprotective effect of isoferulic acid, a polyphenol which was identified in an extract of foxtail millet bran. Experiments were carried out in vitro in human gastric epithelial cells GES-1, and in an animal model of ulcer (ethanol-induced gastric mucosal injury). The role of GALNT2 was investigated as well.

Several points need to be elucidated and/or modified in the manuscript. The introduction section should be expanded reporting other dietary polyphenols with gastroprotective activity in this model, and their natural sources. This information will strengthen the role of such dietary components in preventing or treating gastric ailments.

Table 1 is not suitable in the introduction and should be moved in supplementary materials section.

The authors must explain why they decided to separate the BPIS in >200 MW and <200 MW. This is not clearly reported in the paper. A bio-guided fractionation is generally used to identify active components in botanicals with biological activities.

The extraction of BPIS should be briefly described, not just cited the reference 19. Moreover, phytochemistry aimed at characterizing BPIS must be added in the manuscript.

In Table 2 is not clear the meaning of the raw reporting protein name, protein ID etc. (line 147).

In addition, methods regarding calculation of injury index or damages induced in gastric mucosa following ethanol administration are widely known, so data reported in Tables 2 and 3 should be moved to the supplementary material section or deleted.

GES-1 cells are not sold by ATCC, please report the real origin of the cell line.

Another major point affecting the publication of the article is the lack of novelty regarding the most active pure compound. At least another manuscript showed the gastroprotective effect of isoferulic acid; as an example, Mariza Abreu Miranda demonstrated that isoferulic acid (30 mg/Kg) reduced ethanol-induced ulcers in mice, suggesting this pure compound as one among the main contributors to the efficacy of an extract from Solanum cernuum Vell (Jouranl of Ethnopharmacology, 2015). The authors do not cite this paper. They should explain and underline the differences between the two papers.

Conclusion and discussion are brief and not exhaustive with respect to the data reported along the manuscript.

Comments on the Quality of English Language

English language is fine, just minor editing required.

Reviewer 3 Report

Comments and Suggestions for Authors

The study investigates the potential of isoferulic acid (IFA), a polyphenol derived from foxtail millet bran, as a functional food ingredient for ameliorating alcohol-induced gastric mucosal injury.

The researchers separated BPIS by molecular weight and identified IFA as the primary active component through a series of cell experiments.

The ameliorative effect of IFA was demonstrated in an ethanol-induced gastric injury animal model using histological and biochemical analyses.

The study reveals that IFA exerts its gastroprotective effect by regulating the activity of the GALNT2 enzyme, which is crucial for mucin glycosylation. IFA was shown to improve mucin synthesis by preventing the ethanol-induced decrease in GALNT2 activity, as evidenced by enzyme activity measurements and the demonstration of IFA-GALNT2 binding.

This research provides evidence supporting the use of functional foods in the prevention and treatment of gastric diseases and identifies new physiological activities of polyphenols found in grain husks. Further research is required to elucidate the precise mechanism by which IFA-GALNT2 binding affects GALNT2 acetylation and fully understand IFA's mode of action.

In conclusion, this contributes to the development of functional food ingredients, employing systematic experimental methods and presenting results that demonstrate IFA's protective effect against alcohol-induced gastric injury, proposing a novel mechanism of action.

Round 2

Reviewer 1 Report

Comments and Suggestions for Authors

-

Reviewer 2 Report

Comments and Suggestions for Authors

Manuscript has been changed according to my suggestions. Paper can be published in the present form.